# Self-Supervised Visual Acoustic Matching

**Arjun Somayazulu**[1]    **Changan Chen**[1]    **Kristen Grauman**[1,2]
[1]UT Austin    [2]FAIR, Meta

## Abstract

Acoustic matching aims to re-synthesize an audio clip to sound as if it were recorded in a target acoustic environment. Existing methods assume access to paired training data, where the audio is observed in both the source and target environments, but this limits the diversity of training data or requires the use of simulated data or heuristics to create paired samples. We propose a self-supervised approach to visual acoustic matching where training samples include only the target scene image and audio—without acoustically mismatched source audio for reference. Our approach jointly learns to disentangle room acoustics and re-synthesize audio into the target environment, via a conditional GAN framework and a novel metric that quantifies the level of residual acoustic information in the de-biased audio. Training with either in-the-wild web data or simulated data, we demonstrate it outperforms the state-of-the-art on multiple challenging datasets and a wide variety of real-world audio and environments. Project page: `https://vision.cs.utexas.edu/projects/ss_vam`

## 1  Introduction

The acoustic properties of the audio we hear are strongly influenced by the geometry of the room and the materials that make up its surfaces—large, empty rooms with hard surfaces like concrete and glass lead to longer reverberation times, while smaller, more cluttered rooms with soft materials like curtains and carpets will absorb sound waves quickly and produce audio that sounds dull and anechoic. Human perception exploits this acoustic-visual correspondence, and we rely on perceiving natural-sounding audio that is consistent with our environment as we navigate daily life.

Likewise, this phenomenon is important in virtual environments, such as in AR/VR applications. When one hears audio that is acoustically consistent with the virtual environment they are seeing, their brain can better integrate audio and visual information, leading to a more immersive experience. Conversely, when one hears audio that does not match the expected acoustics of the virtual environment, the perceptual mismatch can be jarring. The problem of audio-visual acoustic correspondence extends well beyond AR/VR applications. Film and media production involve recording audio in diverse spaces, which can be expensive and logistically challenging. Similarly, interior designers and architects face the problem of previewing how a space will sound before it is built.

Today's approaches for modeling acoustic-visual coherence typically assume physical access to the target space [40, 25, 5, 43], which can be impractical or impossible in some cases. In particular, in *acoustic matching*, the audio captured in one environment is re-synthesized to sound as if it were recorded in another target environment, by matching the statistics (e.g. reverberation time) of audio samples recorded in that target environment [9, 15, 21, 24, 26, 47]. *Visual acoustic matching* (VAM) instead takes an image of the target space as input, learning to transform the audio to match the likely acoustics in the depicted visual scene [3] (see Figure 1(a)). In both cases, however, learned models ideally have access to *paired* training data, where each training audio clip is recorded in two different environments. This permits a straightforward supervised learning strategy, since a model can learn to transform one clip (source) to the other (target). See Figure 1(b), left. Unfortunately, this approach

37th Conference on Neural Information Processing Systems (NeurIPS 2023).

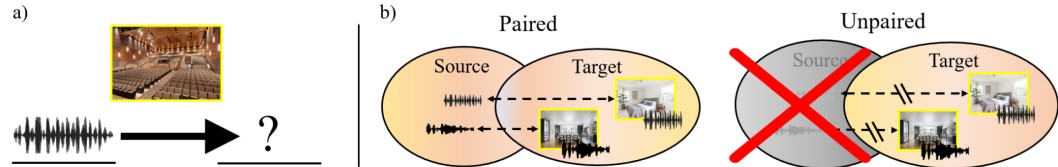

Figure 1: **Self-supervised visual acoustic matching.** (a) Given source audio and target image[1], the goal is to re-synthesize the audio to reflect the acoustics of the target environment. b) Two possible data settings: the *paired audio* setting (left) observes both the source audio and the audio in the target environment, allowing for supervised training, while the *unpaired audio* setting (right), observes only the audio in the target environment. Our self-supervised strategy handles the unpaired setting.

puts heavy demands on data collection: large-scale collection of paired data from a variety of diverse environments is typically impractical.

In-the-wild web videos provide us with a large, readily available corpus of diverse acoustic environments and human speakers. However, this data is *unpaired*—we only observe the sound recorded in the target space, without a second recording in a different environment for reference. Prior work attempts to turn unpaired data into paired data by using an off-the-shelf dereverberator model [7] to produce pseudo-anechoic source recordings from in-the-wild reverberant audio, which are then passed to a visual acoustic matching (VAM) model with the true reverberant audio as the target [3]. While that approach is inspiring, it has a fundamental limitation. The automatic dereverberation process is necessarily imperfect, which means that *residual acoustic information indicative of the target environment's acoustics can remain in the (pseudo) source example*. In turn, the acoustic matching model trained to produce the target audio can learn to use those residual acoustic cues in the audio—instead of the target space's visual features. When evaluated at test-time on arbitrary source audio and unseen images, the residual acoustic clues exploited during training are no longer available, leading to poor acoustically matched audio outputs.

We propose a *self-supervised visual acoustic matching* approach that accommodates training with unpaired data from in-the-wild videos (See Figure 1(b), right). Our key insight is a training objective that explicitly removes residual acoustics in the audio, forcing reliance on the visual target. In particular, our approach jointly trains an audio-visual *debiaser*—which is trained to adversarially minimize residual acoustic information in the dereverberated audio—alongside a reverberator that performs visual acoustic matching. To this end, we develop an *acoustic residue* metric that quantifies the level of residual acoustic information in a waveform, based on the difference in performance between a) an acoustic matching model that conditions on the target image and b) a model that does not condition on any image. Intuitively, training on audio with low residual acoustic information frees the model from relying on that (unrealistic) information during training, allowing it to instead leverage the necessary visual cues.

We use a time-domain conditional GAN as our debiaser, and continually update the integrated reverberator as the distribution of generated audio shifts during training. Unlike prior work, our approach allows training directly on unpaired videos and speech.[2]

Our proposed LeMARA model outperforms existing approaches [3, 7, 14] on challenging in-the-wild audio and environments from multiple datasets. Further, to benchmark this task, we introduce a high audio-visual correspondence subset of the AVSpeech [10] video dataset. Though we focus on the task of visual-acoustic matching, our insight potentially has broader implications for other self-supervised multi-modal learning tasks in which one wants to control the impact of a dominating modality.

## 2 Related work

**Room acoustics and spatial sound** Audio-visual acoustic matching has limited prior work, though there is growing interest from vision and audio communities. Image2Reverb [40] learns to map an input image to its corresponding Room Impulse Response (RIR)—the transfer function that

---

[1] `https://rulonco.com/features-modern-auditorium-design/`

[2] We focus on clean human speech in indoor settings given its relevance to many of the applications cited above, and due to the fact that human listeners have strong prior knowledge about how room acoustics should affect speech. However, our model design is not specific to speech and could be applied to other audio sources.

characterizes acoustics at a particular listener/speaker location—which can then be convolved with an arbitrary source waveform. In audio-only approaches, [41, 32, 33] generate RIRs from reverberant audio clips, and [42] uses a WaveNet model to warp source audio into a target space conditioned on a learned embedding of the target audio. Generalizing RIR inference to full 3D environments, Few-Shot RIR [25] and Neural Acoustic Fields [23] sample impulse responses at multiple places in an environment in order to synthesize the RIR at novel locations with a transformer. In related tasks, Novel View Acoustic Synthesis [5] directly synthesizes a source audio at a new camera pose in the room, while other methods binauralize monaural source audio [16, 36]. Unlike our model, which can train from arbitrary images, existing methods require knowledge of the ground truth RIRs [41, 32, 33, 40, 23, 25] or paired source-target audio [5, 16, 36]. Most relevant to our approach, AViTAR [3] uses a cross-modal transformer to re-synthesize the audio; it relies on an off-the-shelf audio-visual dereverberator trained in simulation [7] to produce a pseudo-clean source signal, and suffers from the acoustic residue issue discussed above. Our results illustrate how our model overcomes this shortcoming.

**Speech synthesis and enhancement**    Recent work for speech re-synthesis treats acoustic properties as a "style" which can be disentangled from underlying speech and used to perform acoustic-style matching of audio [27, 31, 30, 46]. However, they require either paired data or speaker embeddings to be learned. Web videos (our target domain) can consist of entirely unique speakers, making it difficult to learn a robust speaker embedding. While environment-aware text-to-speech [43] is applicable even in settings where each speaker in the dataset is unique, the model requires access to paired speech for supervised training. Supervised methods for speech enhancement and dereverberation assume access to paired training data, i.e., anechoic "clean" reference waveforms alongside the altered waveforms [44, 7, 48, 37, 1, 12, 13]. Unsupervised speech enhancement approaches [22, 19, 14] such as MetricGAN-U [14] optimize generic perceptual speech quality scores, such as PESQ [38] (which requires paired samples) and speech-to-reverberation modulation energy (SRMR) [11] (which does not). While we share the objective of relaxing supervision requirements, our overall goal is distinct: rather than optimize a generic quality metric, we aim to retarget the input sound to match a specific (pictured) environment. To achieve that goal, we introduce a novel debiasing objective applicable in a conditional GAN framework.

## 3   Approach

We present **Le**arning to **M**atch **A**coustics by **R**emoving **A**coustics, **LeMARA**, to achieve self-supervised visual acoustic matching (VAM). Let $A \in \mathcal{A}$ denote an audio waveform and let $V \in \mathcal{V}$ denote an image frame. During training, we are given $N$ unlabeled examples of *target* audio and scenes $\{(A_t, V_t)\}_{t=1}^N$, taken from frames and accompanying audio in YouTube videos (cf. Sec. 4 for dataset details). While the data is multi-modal, it is unpaired: it has both audio and visual features, but it lacks a paired sample of the audio in some other source domain (see Fig. 1). With this data, we wish to learn a function $f(A_s, V_t) : \mathcal{A}, \mathcal{V} \rightarrow \mathcal{A}$ that takes *source* audio $A_s$ (which may or may not be reverberant) and a *target* image $V_t$, and produces the audio re-synthesized to sound like it was recorded in the target scene.

To self-supervise $f$ using unpaired training data, we would like a model that (1) dereverberates $A_t$ to strip it of its room acoustics, yielding pseudo-source audio $\hat{A}_t^{(s)}$ and then (2) reverberates $\hat{A}_t^{(s)}$ by "adding" in the room acoustics of image $V_t$ to regenerate the target audio: $f(\hat{A}_t^{(s)}, V_t) \approx A_t$. A naive joint training of two such modules—a dereverberator and reverberator—would collapse to a trivial solution of $f$ doing nothing. A better solution would pre-train the dereverberator with a well-trained audio-only model, yet as we show in experiments, this leaves residuals of the target environment in the audio $\hat{A}_t^{(s)}$ that handicap the reverberator at inference time, when such signals will not exist.

The key insight of our approach is to make the dereverberation stage a *de-biasing* stage that is explicitly trained to remove room acoustics information not available in the target image $V_t$. This forces reliance on the visual target and helps our model $f$ generalize to unseen audio and scenes.

Our model has two main components that are trained jointly: (1) a de-biaser model $G$ responsible for removing room acoustics from the target audio and (2) a visually-guided reverberator $R$ that injects acoustics from the target environment into the output waveform. We first introduce the de-biaser architecture (Sec. 3.1), followed by our novel acoustic residue metric that is optimized during de-biaser training (Sec. 3.2), and our training strategy that allows joint fine-tuning of the

reverberator and de-biaser (Sec. 3.3). Finally, we present our approach for training self-supervised VAM (Sec. 3.4). Figure 3 overviews our model.

## 3.1  De-biasing Conditional Generative Adversarial Network

The role of our de-biaser is to dereverberate audio samples in a way that minimizes any residual room acoustics information. To that end, we devise an adversarial de-biaser based on MetricGANs [12, 14]. MetricGANs are similar to conventional generative adversarial networks (GAN) [17]—with a generator that aims to enhance a speech waveform—except the discriminator's job is to mimic the behavior of some target *quality function*. Our de-biaser module extends the basic MetricGAN, augmenting it with novel acoustic residue metric (Sec. 3.2) and training procedure (Sec. 3.3) that accounts for the evolving distribution of the de-biased audio.

Our GAN consists of a generator $G$, a discriminator $D$, and an audio quality metric $\mathcal{M}$. $D$ is trained to approximate the quality metric $\mathcal{M}$, and $G$ is trained to maximize the metric score on generated data, using $D$ as a learned surrogate of the metric function $\mathcal{M}$. $G$ is a conditional generator: given an input waveform, it produces a modified waveform which optimizes the quality metric score.

Let $\{A_t, V_t\}$ be a dataset of target audio-image pairs, and let $\mathcal{M}(A_t, V_t) \in [0, 1]$ be our quality metric $\mathcal{M}$ (defined below) that produces a scalar measure of the residual room acoustic information in audio. As in the conventional GAN framework, we alternate between discriminator and generator updates. During an epoch of discriminator training, $D$ trains to approximate the metric function $\mathcal{M}$'s scores on both raw audio $A_t$ and generated audio $G(A_t)$. The discriminator loss function is:

$$\mathcal{L}_D = \|D(A_t) - \mathcal{M}(A_t, V_t)\|_2 + \|D(G(A_t)) - \mathcal{M}(G(A_t), V_t)\|_2 + \|D(A_{hist}) - s_{hist}\|_2, \quad (1)$$

where the first and second terms incentivize $D$ to produce score estimates that approximate the metric function when evaluated on raw input audio ($A_t$) and generated audio $G(A_t)$, respectively. Following [12], the third term trains the discriminator on samples from a historical replay buffer $\{(A_{hist}, s_{hist})\}$, where $A_{hist} = G_{prev}(A_t)$ is a generated sample from a previous epoch, and $s_{hist} = \mathcal{M}_{prev}(G_{prev}(A_t), V_t)$ is its associated metric score. Training on these historical samples helps improve stability and mitigates issues with catastrophic forgetting in the discriminator.

The generator is trained with an adversarial loss, using the discriminator $D$ learned from the previous epoch of discriminator training (which depends only on $A_t$) as a surrogate of the true metric $\mathcal{M}$ (which depends on both $A_t$ and $V_t$). The generator loss is

$$\mathcal{L}_G = \|D(G(A_t)) - 1\|_2. \quad (2)$$

Our metric is normalized to produce scores between 0 and 1 (1 being optimal), so this loss forces $G$ to generate audio that maximizes the estimated metric score. Next, we introduce our metric $\mathcal{M}$.

## 3.2  Acoustic Residue Metric

Rather than train the de-biaser GAN to optimize a generic speech quality metric [12, 14], we wish to quantify the amount of *residual room acoustics information* in an audio sample. Hence, we define a metric $\mathcal{M}$ that allows the downstream reverberator model $R$ itself to quantify the level of residual acoustic information in the waveform. Specifically, the metric consists of two models trained to perform reverberation on dereverberated speech. Importantly, one model $R_v$ has been trained to perform VAM (using the target image as conditioner), while the other, $R_b$, has been trained to perform blind acoustic matching (without the target image as conditioner). We next define the reverberator modules, and then return to their role in $\mathcal{M}$.

Inspired by recent work in time-domain signal modeling [45, 36, 5], we use a WaveNet-like architecture for the reverberators consisting of multiple stacked blocks of 1D convolutional layers, with an optional gated fusion network to inject visual information for $R_v$. Similar to [5], the reverberators use a sinusoidal activation function followed by two separate 1D conv layers that produce residual and skip connections, the latter being mean pooled and fed to a decoder to produce reverberated audio. We choose this model because it is parameter-efficient, consisting entirely of 1D convolutions, and because it allows time-domain conditional waveform generation. See Sec. 5 for training details.

We use these models to compute the acoustic residue metric. Given input audio $A$ and image $V$, our metric is defined as:

$$\mathcal{M}(A, V) = \sigma \left( \frac{|\mathcal{RT}(R_b(A)) - \mathcal{RT}(A_t)| - |\mathcal{RT}(R_v(A, V)) - \mathcal{RT}(A_t)|}{\max(0.1, \mathcal{RT}(A_t))} \right), \quad (3)$$

where $A_t$ denotes the known target audio and $\mathcal{RT}$ is a scalar-output function characterizing the general reverberant properties of its input audio, which we define below.

Eqn. 3 quantifies the level of acoustic residue—that is, how much greater the blind reverberator's error is compared to the visually-guided reverberator's error. If de-biasing of $A$ has gone well, this value will be high. When evaluated on audio that contains a high level of residual acoustic information, the visual features will not provide additional useful information, resulting in similar performance by both visual and blind reverberation models. In other words, if the two errors are similar, visual is not adding much, and there is room acoustic information lingering in the audio $A$. This pushes the $\mathcal{M}$ score to be smaller

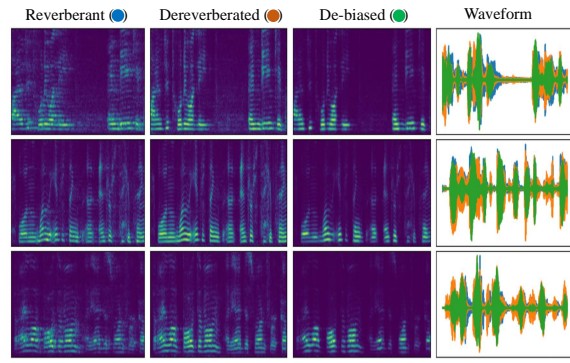

Figure 2: **De-biased audio.** The de-biaser removes residual acoustic traces in audio, and attenuates long sound decay faster than in dereverberated speech.

(poor quality under the metric). On the other hand, when a waveform contains very little residual acoustic information, the visual features will help the visual model $R_v$ produce a more accurate acoustically matched waveform model with lower error compared to the blind model $R_b$. This will result in a higher metric score.

Figure 2 demonstrates the effect of de-biasing. Reverberant audio (left) has been imperfectly dereverberated (middle), leaving residual reverberation trails which contain information about the original acoustic environment. De-biased audio (right) removes these residual artifacts. The waveform plots (right) show de-biased audio (green) significantly attenuates the long sound decay present in both reverberant (blue) and dereverberated (orange) audio. This forces the reverberator to learn acoustics from the target image.

For the function $\mathcal{RT}$ in Eqn. 3, we leverage a classic content-invariant metric for characterizing room acoustics: the Reverberation Time at 60 or "RT60". RT60 is the amount of time after a sound source ceases that it takes for the sound pressure level to reduce by 60 dB—which depends directly on the geometry and materials of the space. For example, no matter the initial direct sound, a big cathedral will yield a high RT60, and a cozy living room will yield a low RT60. While RT60 can be quantified by sensing when one has access to the physical environment, we use a learned estimator for RT60 to allow generalization (see Sec. 5 for details). In Eqn. 3, the normalization by the RT60 of the target audio improves stability of discriminator training. We use the clipping function $max(0.1, \cdot)$ here to prevent samples with extremely low RT60 from destabilizing training.

Training with this acoustic residue metric allows the downstream reverberator models $R_v, R_b$ themselves to improve the performance of the de-biaser model $G$. Unlike SRMR, DNSMOS [35], or any existing off-the-shelf metric that quantifies general speech quality, our metric directly addresses the problem of *residual* acoustic information in audio. Although $G$ may learn a function similar to that of dereverberation, we use the term de-biaser to describe the generator to highlight the novel training objective it is trained against, which distinguishes it from a conventional dereverberation model.

### 3.3    Joint Training of the De-biaser and Reverberators

At initialization, $R_v$ and $R_b$ are trained on a certain distribution of speech. When training the de-biasing GAN with the acoustic residue metric, generated speech can eventually fall out of the distribution on which these reverberators were trained, causing $\mathcal{M}$ to produce unreliable metric scores that destabilize training. To address this, we introduce a strategy to update $R_v$ and $R_b$ during training of the de-biasing GAN. Updating these models ensures that $\mathcal{M}$ consistently produces reliable acoustic residue scores as the distribution of generated speech shifts over the course of GAN training.

At the start of GAN training , we initialize the *target networks* $R_v^t$ and $R_b^t$ as copies of $R_v$ and $R_b$ respectively. During discriminator training, each batch of $\{(G(A_i), V_i)\}$ samples are passed to $R_v$ and $R_b$ to compute metric scores under their current frozen state. This batch is also passed to the

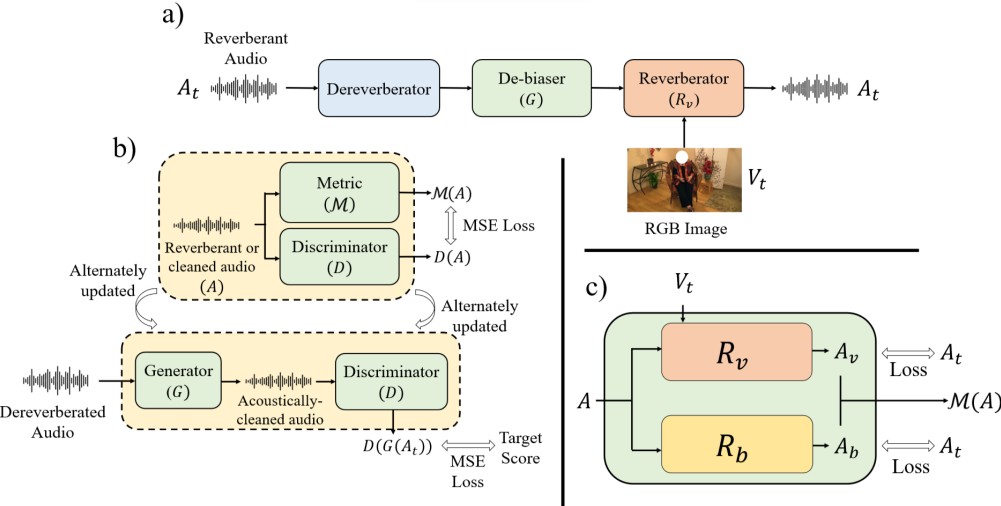

Figure 3: **LeMARA overview. a) Training procedure**. Reverberant audio is first processed with an off-the-shelf dereverberator. It is then input to a de-biaser model which strips acoustics from the audio. The clean audio is passed to the reverberator along with the target image for acoustic matching. **b) De-biaser architecture**. $G$ is trained to adversarially maximize the score of the discriminator $D$, which learns a surrogate of the acoustic residue metric $\mathcal{M}$ (Sec. 3.1 and 3.2). **c) Acoustic residue metric**. Both $R_v$ and $R_b$ are continually trained on generated data to ensure that they provide accurate metric scores as the distribution of generated data evolves during training (Sec. 3.3). **At test time**, we use the trained de-biaser $G$ and the visual reverberator $R_v$ to perform VAM.

target networks, which compute the losses

$$\mathcal{L}_{\text{visual}} = ||R_v^t(G(A), V) - A||_2 \tag{4}$$

$$\mathcal{L}_{\text{blind}} = ||R_b^t(G(A)) - A||_2 \tag{5}$$

and perform an update step. Every $E$ epochs, the target networks' weights are copied over into the metric networks. This update strategy allows $R_v$ and $R_b$ to be jointly fine-tuned with the de-biaser model $G$. Figure 3 overviews the model components and data flow.

### 3.4 Training and Inference

Training proceeds in three steps. (1) First we pretrain the de-biaser $G$. This entails pretraining a MetricGAN-U [14] with the Speech-to-Energy-Modulation-Ratio (SRMR) metric [11] on speech pre-processed with our off-the-shelf dereverberator (see Sec. 5). By refining the dereverberated output with the MetricGAN-U, we improve its quality and intelligibility without introducing additional supervision requirements. (2) Second, we pretrain the reverberators that perform (visual) acoustic matching. Specifically, we train $R_v$ and $R_b$ with $\mathcal{L}_{visual}$ and $\mathcal{L}_{blind}$, respectively, using the dereverberated and SRMR-optimized outputs from the MetricGAN in step (1). (3) Finally, we jointly fine-tune both the de-biaser $G$ and reverberators, using the acoustic residue metric (Eqn. 3) for the GAN metric $\mathcal{M}$, the generator and discriminator losses given in Eqns. 2, and 1 together with our alternating training scheme defined in Sec. 3.3. To improve stability in training, since the discriminator $D$ starts with a good approximation of the SRMR metric, we continue training in step 3 using a weighted combination of SRMR and our residue metric: $\alpha\text{SRMR}(A) + (1 - \alpha)\mathcal{M}(A, V)$. Although the acoustic residue is approximately differentiable, the implementation of SRMR we use is not, motivating our use of MetricGAN over an end-to-end alternative. This approach also allows for extensibility to other non-differentiable speech quality scores such as PESQ.

At test time, we use LeMARA for visual-acoustic matching as follows: given a source audio $A_s^{(q)}$ and target environment image $V_t^{(q)}$, we apply the trained de-biaser $G$ followed by the visual reverberator $R_v$:

$$f(A_s^{(q)}, V_t^{(q)}) = R_v(G(A_s^{(q)}), V_t^{(q)}). \tag{6}$$

Altogether, our approach adds the room acoustics depicted in $V_t^{(q)}$ to the input audio. In the case that the source audio $A_s^{(q)}$ is known to be anechoic (e.g., a user is using LeMARA to re-synthesize stock anechoic sounds for a new scene), then we simply bypass the de-biaser $G$ and directly apply $R_v$.

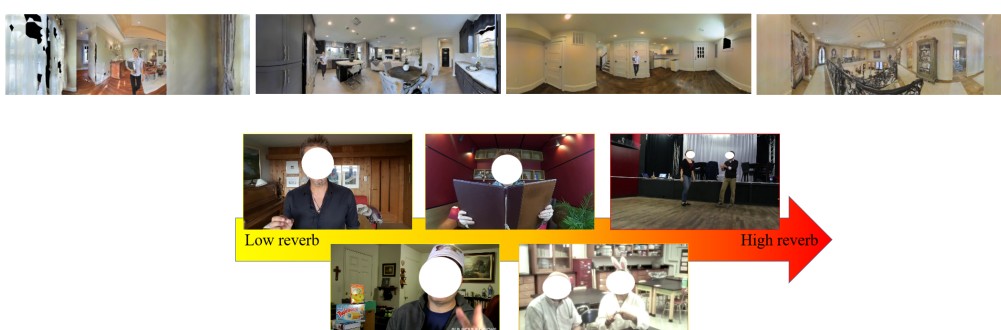

Figure 4: **Datasets. SoundSpaces-Speech** (top row) renders panoramic views of people in indoor environments. **AVSpeech-Rooms** (bottom) contains a wide variety of naturalistic indoor environments with diverse acoustic properties.

## 4 Datasets

We use two datasets: SoundSpaces-Speech [7] and AVSpeech [10]. See Figure 4. For all results, we test only on audio and environments that are not observed during training.

**SoundSpaces-Speech** SoundSpaces-Speech [7] is an audio-visual dataset created using the SoundSpaces audio simulation platform [4, 6] together with 82 3D home scans from Matterport3D [2] and anechoic speech samples from LibriSpeech [28]. SoundSpaces offers state-of-the-art generation of Room Impulse Responses (RIRs) computed at regular source-listener locations throughout the environment, accounting for all major real-world acoustic phenomena (reverberation, early specular/diffuse reflections, etc.) See [7] for details.

By convolving an anechoic sample with the appropriate RIR, it allows realistic simulation of audio reverberation at arbitrary locations in the pre-scanned real-world Matterport environments. A 3D humanoid is rendered at the source location in the Matterport home. SoundSpaces-Speech consists of anechoic speech samples from LibriSpeech paired with their acoustically-correct reverberated waveform (rendered using SoundSpaces) in any of 82 unique environments, together with an RGB-D image at the listener's position. To train on SoundSpaces-Speech in a self-supervised setting, we discard the source audio and only use the target audio (simulated reverberant audio). We use train/val/test splits of 28,853/280/1,489 samples.

**AVSpeech-Rooms** AVSpeech [10] is a large-scale video dataset consisting of 3-10 second clips from YouTube videos, most of which feature a single speaker and little background noise. Not all of the clips have naturalistic audio-visual correpondence, due to video editing tricks, microphone locations, virtual backgrounds, etc. Hence, we create a subset of AVSpeech, called AVSpeech-Rooms, that preserves samples with useful information about room geometry and material types (see Supp. for details). A randomly selected frame from the video clip is used as the target image. Our final set consists of 72,615/1,911/1,911 train/val/test samples. See Figure 4 (bottom).

## 5 Experiments

**Implementation Details** We use the procedure outlined in Sec. 3.4 to train on SoundSpaces-Speech. For training on AVSpeech-Rooms, we proceed directly to step (2), pre-training $R_v$ and $R_b$ on audio that has been de-biased using the fine-tuned SoundSpaces-Speech de-biaser model (instead of an SRMR-optimized model trained on AVSpeech-Rooms). We then proceed to step (3) as in SoundSpaces-Speech. We refer to this setup as "shortcut training" to highlight our use of the SoundSpaces-Speech trained de-biaser to bypass step (1) when training on AVSpeech-Rooms. While AVSpeech can be trained with the full procedure outlined in Sec. 3.4 (see ablations in Supp.), shortcut training allows us the advantage of the strong prior for de-biasing learned by the fine-tuned SoundSpaces de-biaser.

We train LeMARA using the combined acoustic residue metric with $\alpha = 0.7$. We train a WaveNet-based dereverberator [36] ("off-the-shelf") on paired SoundSpaces-Speech audio which is "reversed" (reverberant input audio, anechoic target audio). We use this dereverberator for both LeMARA and the ViGAS [5] baseline. Prior VAM work [3] used an audio-visual dereverberator [7] trained on both

Table 1: VAM results on two datasets (unseen environments). RTE standard errors are less than 0.005, and STFT standard errors are 0.031 and 0.64 for the two datasets, respectively.

| Train
Test
Model | *SoundSpaces-Speech*
*SoundSpaces-Speech* | | | *AVSpeech-Rooms*
*AVSpeech-Rooms* | | | *LibriSpeech* |
|---|---|---|---|---|---|---|---|
| | RTE (s) | STFT | logSTFT | RTE (s) | STFT | logSTFT | RTE (s) |
| Input audio | 0.320 | 1.427 | 1.274 | 0.310 | **1.327** | 2.107 | 0.358 |
| AViTAR | 0.080 | 2.471 | 1.629 | 0.136 | 2.894 | 1.290 | 0.239 |
| ViGAS | 0.108 | 4.373 | **1.232** | 0.109 | 7.007 | 0.662 | 0.254 |
| LeMARA(no vis) | 0.152 | 5.612 | 1.884 | 0.137 | 6.256 | 0.705 | 0.223 |
| LeMARA (ours) | **0.079** | **0.690** | 1.530 | **0.071** | 6.298 | **0.571** | **0.210** |

simulated and real-world data to pre-process reverberant audio. For fair evaluation, we train their model with their same original audio-visual dereverberator.[3]

We adapt our code for the reverberator models and ViGAS from [36].[4]. We train ViGAS with the same hyperparameters and loss as our reverberators during pre-training. Our de-biaser is adapted from the speechbrain MetricGAN-U implementation [34]. Our RT60 estimator is trained on pairs of reverberant samples from a SoTA audio simulator [6] paired with the ground truth RT60 for the RIR used to produce the reverberant sample. We use a Resnet18 [18] model to encode our visual features on RGB images. The last feature map before the final pooling layer is flattened and used as the visual feature conditioner. See Supp. for training details and architecture for these models. We plan to release our code to facilitate further research.

**Metrics**   We rely on three metrics to evaluate quality of VAM models. **STFT Error** and **logSTFT Error** compute the MSE loss and log MSE loss, respectively, between the magnitude spectrograms of predicted and target speech. Because we do not have access to the ground truth RIR, we also use an RIR reverberation metric that can be reliably estimated from audio, **RT60 Error (RTE)**, which measures the MSE between RT60 estimates of predicted and target speech. The first two apply only when we have ground truth re-synthesized audio (in simulation), while the third is content-invariant and captures room acoustics signatures for any target.

**Seen and unseen evaluation**   We report results on two different settings: *unseen environments*, where both the source audio $A_s$ and target image $V_t$ come from the test set; and *seen environments*, where an audio sample $A_s$ from the test set is paired with a target image $V_t$ from the train set (seen by the model). The unseen environment setting is important for evaluating generalization to novel scenes (e.g. matching an audio sample to an image from the Web), while the seen environment setting is useful for cases in which we already have video recordings, such as in film dubbing.

**Baselines**   As baselines, we compare to state-of-the-art models for audio-visual re-targeting: (1) **AViTAR** [3], the only prior method that addresses the visual-acoustic matching task. It consists of a Transformer for audio-visual fusion, followed by a generator that synthesizes the reverberant waveform from the audio-visual latent feature. As discussed above, for self-supervised training, the authors use a pre-trained audio-visual dereverberation model [7] to create pseudo-source audio, which is passed as input to the model. (2) **ViGAS** [5], a model designed for novel-view acoustic synthesis, conditioned on a camera pose. We adopt its Acoustic Synthesis module, a WaveNet model based on [36], for our task. To apply it for VAM, we replace the camera pose with the flattened feature from the ResNet. (3) **Non-visual LeMARA**. We evaluate LeMARA with the blind reverberator $R_b$ fine-tuned during training. (4) **Input audio**. We copy the dereverberated audio to the output. AViTAR and ViGAS are trained with the data augmentation strategy introduced in [3] (see Supp. for details).

**Results on SoundSpaces-Speech**   Tables 1 and 2 report results on SoundSpaces-Speech (left three columns) for the unseen and seen settings, respectively. LeMARA outperforms the baselines on most metrics. Non-visual LeMARA performs significantly worse, indicating LeMARA effectively utilizes visual features during acoustic matching. This demonstrates the success of our acoustic de-biasing, which forces stronger learning from the visual stream. Notably, our model significantly outperforms

---

[3] https://github.com/facebookresearch/visual-acoustic-matching
[4] https://github.com/facebookresearch/BinauralSpeechSynthesis

Table 2: Results on seen environments. We evaluate VAM using audio samples from the test set paired with target images from the train set.

| Model | SoundSpaces-Speech | | | AVSpeech-Rooms | | |
|---|---|---|---|---|---|---|
| | RTE (s) | STFT | logSTFT | RTE (s) | STFT | logSTFT |
| AViTAR | 0.062 | 3.186 | 1.506 | 0.131 | **2.852** | 1.273 |
| ViGAS | 0.076 | 6.386 | **0.962** | 0.111 | 6.946 | 0.657 |
| LeMARA (ours) | **0.060** | **0.667** | 1.417 | **0.067** | 6.198 | **0.570** |

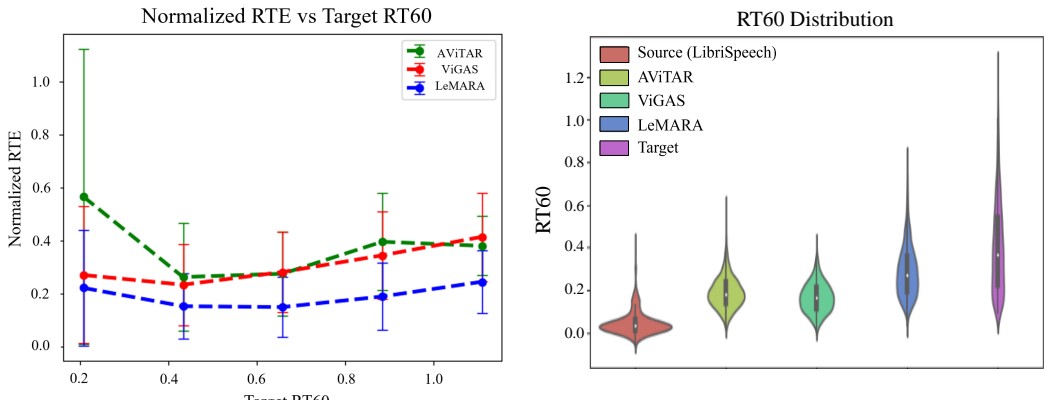

Figure 5: **Normalized RTE across reverberation levels (left).** We evaluate RTE normalized by target RT60 on AVSpeech-Rooms samples with varying levels of reverberation. LeMARA consistently achieves lowest error at all reverberation levels. **RT60 distribution (right).** The distribution of RT60 values for audio generated by LeMARA best matches the target distribution.

ViGAS—which shares the same architecture as our reverberator —indicating that our performance improvement over AViTAR can be attributed to our novel training objective, and not simply due to a shift in architecture from Transformer to WaveNet.

**Results on AVSpeech-Rooms**  Tables 1 and  2 show results on AVSpeech-Rooms (right four columns) for both unseen and seen settings, respectively. We further divide the unseen evaluation into two scenarios: (1) Where the source audio and target visual come from the same (unobserved) AVSpeech-Rooms sample and (2) where the source audio comes from LibriSpeech [29], a dataset of anechoic source samples of people reading passages in English (rightmost column). In both cases the target visual environment is a frame from an unseen AVSpeech video.

LeMARA outperforms the baselines in both settings on RTE. We outperform ViGAS in the LibriSpeech scenario despite sharing the same reverberator architecture, highlighting the efficacy of our novel training objective. Figure 5 (right) shows the distribution of RT60 values for audio reverberated by our model (blue), the baselines, and the target RT60 distribution (purple). Our model best matches the target RT60 distribution. LeMARA also outperforms baselines across samples with a diverse range of real-world reverberation intensities. We stratify the test set by ground truth RT60 and evaluate performance on samples within each reverberation bin (Figure 5, left). LeMARA outperforms the baselines across all reverberation levels, demonstrating its robustness to variation in real-world data.

Although our model performs poorly on STFT error, we significantly outperform existing approaches on logSTFT error, which better reflects perceptual quality given the logarithmic nature of human hearing. Furthermore, the naive baseline achieves the lowest STFT error, indicating that dereverberated audio itself strongly resembles reverberant audio even prior to acoustic matching. Models that use this dereverberator without further de-biasing will display artificially low STFT error when evaluated in-dataset (AVSpeech-Rooms → AVSpeech-Rooms). Thus, it is important to balance the in-dataset evaluation with the LibriSpeech generalization case (far right in Table 1, where we soundly outperform Input audio) to gain a complete picture of model performance.

Furthermore, AVSpeech-Rooms contains a variety of non-speech sounds (e.g. air conditioning, clicking/tapping noises from object interactions) which make reverberation even more challenging. These signals may be perceptually weak, but they appear on spectrograms and contribute to the larger

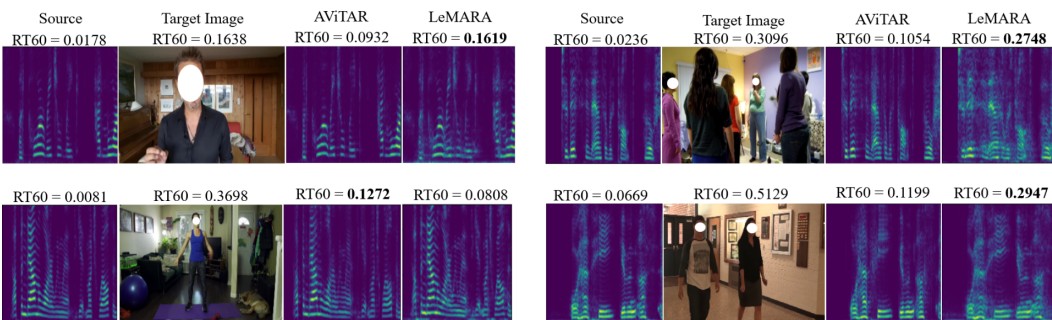

Figure 6: **Acoustically matched audio.** LeMARA produces audio with more accurate acoustics than AViTAR [3] across diverse acoustic and visual scenes. Shown here for LibriSpeech setting.

STFT error we observe on AVSpeech-Rooms. In contrast, SoundSpaces-Speech contains no artifacts or background noise that could contribute to spectrogram errors.

An ablation study of our model analyzing the impact of varying the target metric during training shows the clear advantage of our residue metric compared to SRMR alone (see Supp.). In particular, training with an SRMR objective alone yields an RTE of 0.2308 on LibriSpeech, whereas training with a pure acoustic residue metric yields an RTE of 0.2156; our combined metric yields an RTE of 0.2123. This reinforces the efficacy of our metric as a training objective: de-biasing is distinct from generic enhancement.

Figure 6 shows examples of our model's generated sounds for different target images, compared to the output of AViTAR [3] (the best performing baseline). We display the RT60 of the source audio, the visual scene's ground truth RT60, and the RT60 of the two method's generated audio. In the majority of cases, our model produces audio that more closely matches the true acoustics of the visual scene. Our model learns to add little reverberation in enclosed environments with soft surfaces (top left), and to add more reverberation in open acoustic environments with hard surfaces (bottom right). We also highlight a failure mode in which our model does not inject proper acoustics (bottom left), likely due to the irregular shape of the room. See our project webpage to listen to examples.

**Human perception study** We augment these quantitative results with a human subject study, in order to gauge whether listeners perceive our results as successfully retargeting the audio to the pictured environment. We design a survey using 23 images $V_t^{(q)}$ selected from AVSpeech-Rooms that show room geometry and materials clearly, and are representative of a diverse variety of acoustic environments. We couple those with 23 anechoic source samples $A_s^{(q)}$ from LibriSpeech [29]. For each sample, we generate the acoustically matched audio with both LeMARA and AViTAR [3]—the best performing baseline. We anonymize and shuffle the generated audio, and ask 10 subjects with normal hearing to identify which room matches best with the audio among three given rooms, one of which is the true room $V_t^{(q)}$. Users correctly identified the target room with 46.1% accuracy on speech generated by LeMARA versus 34.7% accuracy with speech generated by AViTAR. This shows our model achieves higher quality acoustic matching according to human perception. That said, the subjects' accuracy rates are fairly low in the absolute sense, which suggests both the difficulty of the problem and subtlety of the perception task. Our results have pushed the state of the art, both objectively and subjectively, but there remain challenges to fully mature visual acoustic matching.

## 6 Conclusion

We introduced a self-supervised approach to visual acoustic matching. Built on a novel idea for disentangling room acoustics from audio with a GAN-debiaser, our model improves the state of the art on two datasets. Our acoustic residue metric and adversarial training has potential to generalize to other multi-modal learning settings where there is risk of unintentionally silencing a paired modality during training. For example, our framework could be explored for binauralization of mono sounds using video or audio-visual source separation. In future work, we plan to explore generalizations to spatial dynamics that would account for movement of a speaker throughout 3D space.

**Acknowledgments:** Thanks to Ami Baid for help in data collection. UT Austin is supported in part by the IFML NSF AI Institute. K.G. is paid as a research scientist at Meta.

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

# 7 Supplementary

In this supplementary material we provide the following:

1. A video for qualitative evaluation of our model's performance (7.1)
2. Details regarding AVSpeech-Rooms curation (7.2) (referenced in Sec. 4 of main paper)
3. Details on our ablation study with different metric training objectives (7.3) (referenced in Sec. 5 — "Results on AVSpeech-Rooms" of main paper)
4. A comparison of audio quality and reverberation metrics between reverberant, de-reverberated, and de-biased audio (7.4)
5. A sample survey slide from our human perception study (Figure 7)
6. Model/training details for our RT60 estimator, de-biaser, discriminator, and reverberator (7.5) (referenced in Sec. 5 — "Implementation Details" of main paper)
7. Details on our data augmentation strategy (7.6) (referenced in Sec. 5 — "Baselines" of main paper)
8. A brief discussion of our work's limitations and broader impact (7.7 7.8)
9. A tabular description of our three-stage training strategy (Table 8)
10. Pseudocode for a discriminator training epoch detailing our reverberator update mechanism (Algorithm 1)
11. A visualization illustrating the effect of speaker distance on the Direct-to-Reverberant Energy Ratio (DRR) in SoundSpaces-Speech audio samples (Figure 9)
12. A visualization of the effect of speaker distance on STFT Error in SoundSpaces-Speech audio samples (Figure 10)
13. A visualization comparing SRMR scores of de-biased and dereverberated data across various reverberation levels for both AVSpeech-Rooms and SoundSpaces-Speech (Figure 11)

## 7.1 Supplementary video

Our video contains several illustrative examples generated by LeMARA on both SoundSpaces-Speech and AVSpeech-Rooms. We provide audio generated by the current state-of-the-art (AViTAR) for reference on each example. We recommend wearing headphones for a better listening experience.

## 7.2 AVSpeech-Rooms

Acoustic AVSpeech consists of audio clips from YouTube videos along with an RGB image frame selected randomly from the corresponding video clip. To create AVSpeech-Rooms, we design a set of criteria which we use to filter out samples in which the image contains uninformative, non-natural, or misleading acoustic information about the space. We focus on cases in which the room is not visible, a microphone is being used, or a virtual background/screen is present — any of which will disturb the natural room acoustics for the speaker's voice. We query each sample with our criteria using a Visual Question Answering (VQA) model [20], which we found more reliable than manual annotations we originally obtained on MTurk. 3 contains information about our criteria.

Table 3: Filtering criteria and % of Acoustic AVSpeech samples removed.

| Question | Answer | dataset % |
|---|---|---|
| Is a microphone or headset visible in the image? | yes | 7.2 |
| Is there a whiteboard/blackboard in the background? | yes | 3.4 |
| Is the entire background one solid color and material? | yes | 23.4 |
| Is there a large projector screen covering most of the background? | yes | 2.2 |
| Is part or all of the background virtual? | yes | 1.3 |
| Are there multiple screens in the image? | yes | 3.5 |
| Is the wider room clearly visible? | no | 3.0 |

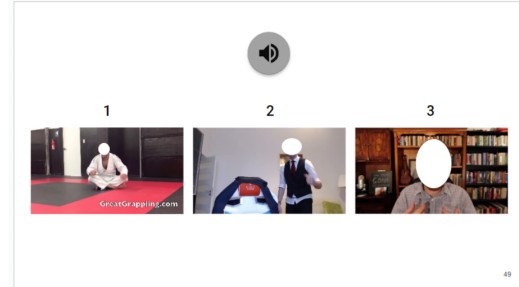

Figure 7: **Human perception study.** The instructions given to the user at the start of the survey (left), and a sample slide from the survey (right). The user is asked to listen to the audio clip, and identify which room image most closely matches its acoustics.

Table 4: Ablation study using different metric training objectives. AR denotes the proposed acoustic residue metric.

| Metric | *LibriSpeech* RTE |
|---|---|
| SRMR [11] | 0.2308 |
| AR | 0.2156 |
| AR (combined) | 0.2123 |
| AR (combined) w/ shortcut | **0.2100** |

## 7.3 Ablations

Table 4 displays our experiments with different self-supervised training objectives. We report performance on the LibriSpeech evaluation setting. The first three rows correspond to experiments in which we do not utilize the shorcut training strategy (referenced in Sec. 3 — "Training" of main paper). Using SRMR alone (row 1) produces the largest (worst) RTE. Training with the acoustic residue metric instead (row 2) leads to a large improvement in RTE, providing empirical support for our metric as an effective training objective. Using our combined metric and the shortcut training strategy (both described in Sec. 3 — "Training" of our main paper) further improves the performance by a small margin.

## 7.4 Evaluation of de-biaser performance

Table 5 evaluates the quality of de-biased audio compared to both reverberant and dereverberated audio. De-biased audio is less reverberant than dereverberated audio as measured by both absolute RT60 metric and RTE (computed against the ground truth anechoic waveform for SoundSpaces-Speech). De-biased audio also achieves a higher SRMR score than dereverberated audio, indicating superior quality and intelligibility.

## 7.5 Model/training details

**RT60 estimator** We adopt the RT60 estimator from [3]. The estimator takes a spectrogram as input, encodes it with a ResNet18 [18], and outputs a scalar RT60 estimate. The model is trained on 2.56s clips of reverberant speech simulated on the SoundSpaces platform [4] paired with the ground truth RT60 computed from the RIR used to generate the reverberant speech. The model trains using MSE loss between predicted and ground truth RT60 values. Ground truth RT60 is computed using the Schroeder method [39].

**De-biaser architecture** The de-biaser $G$ takes a magnitude spectogram as input. This is passed to a bi-directional LSTM with input size 257 and two hidden layers each of size 200, which produces an output with the same temporal length as the input spectrogram. This is passed through a linear layer of size 300 and a leakyReLU activation, followed by another linear layer of size 257 and a Sigmoid activation. The final mask is multiplied with the input magnitude spectrogram to create the generated

Table 5: Evaluation of de-biased audio vs dereverberated audio on various metrics.

| Audio | SoundSpaces-Speech | | | AVSpeech-Rooms | |
|---|---|---|---|---|---|
| | RT60 | RTE | SRMR ($\uparrow$) | RT60 | SRMR ($\uparrow$) |
| Reverberant | 0.40 | 0.36 | 5.97 | 0.42 | 6.76 |
| Dereverberated | 0.06 | 0.02 | 8.35 | 0.06 | 8.99 |
| De-biased | **0.04** | **0.01** | **9.50** | **0.03** | **13.14** |

magnitude spectrogram. A resynthesis module computes phase information from the input audio waveform, combines this with the generated magnitude spectrogram, and performs an inverse STFT to produce the generated waveform. The discriminator $D$ consists of 4 2D Convolutional layers with kernel size (5,5) and 15 output channels, followed by a channel averaging operation and two linear layers of sizes 50 and 10. A LeakyReLU activation with negative slope = 0.3 is used after each intermediate layer. The final layer outputs a scalar-valued metric score estimate.

**De-biaser training**   In stage (1) (see Sec. 3 — "Training" of our main paper), we train with batch size 32. During stage (3) fine-tuning, we use a batch size of 2. $G$ and $D$ are trained with learning rates of 2e-6 and 5e-4 respectively in both stages. In each epoch, We train on 10k samples randomly selected from the train set without replacement. The reverberator models $R_v$ and $R_b$ are updated with the target networks at a frequency of $E = 8$ epochs. For all models, we clip each audio sample to 2.56s during training and evaluation.

**Reverberator training**   We train the reverberators with batch size 4 and a learning rate of 1e-2 in stage (2). During stage (3) fine=tuning, we use batch size 2 and a learning rate of 1e-6. Both reverberator models and the ViGAS baseline are trained with MSE loss between the log magnitude spectrogram of predicted and ground truth audio.

**Baseline training details**   We use a learning rate of 1e-2 and a batch size of 4 to train ViGAS. We train AViTAR with batch size 4 — all other hyperparameters are set as described in [3].

**Compute**   All models are trained on 8 NVIDIA Quadro RTX 6000 GPUs.

## 7.6   Augmentation strategy

We follow a data augmentation strategy similar to that proposed in [3] for training the baseline models, which was shown to produce better generalization performance on the LibriSpeech setting than when trained without this augmentation strategy. In particular, to each batch of dereverberated audio we add colored noise, perform a polarity inversion on the waveform with $p = 0.5$, and convolve the waveform with a randomly selected Room Impulse Response (RIR) from a different acoustic environment with $p = 0.9$. At test time, we evaluate without these audio augmentations. This strategy is designed to mask over residual acoustic information in dereverberated audio during training. We do not use this augmentation strategy in our approach as our model directly learns to remove residual acoustic information, obviating the need for a heuristic strategy to mask it out.

## 7.7   Limitations

Our approach focuses on visual acoustic matching on mono-channel audio exclusively. However, binaural cues in audio play a fundamental role in our perception of reverberation and room acoustics [8]. We leave it to future work to extend our approach to binaural audio.

## 7.8   Broader impact

While training on in-the-wild web videos allows wider access to a diverse variety of speakers and environments, it also introduces uncontrolled biases, speaker privacy concerns, and potentially harmful content into the model.

## 7.9 Data examples

Refer to video to view samples from both SoundSpaces-Speech and AVSpeech-Rooms.

## 7.10 LeMARA training procedure

| | Dataset | |
|---|---|---|
| Train Step # | SoundSpaces-Speech | AVSpeech-Rooms |
| 1 | Pre-train $G$ with SRMR | – |
| 2 | Pre-train $R_v, R_b$ | Pre-train $R_v, R_b$ |
| 3 | Train $G, R_v, R_b$ | Train $G, R_v, R_b$ |

Figure 8: **LeMARA three-step training.** Refer to Section 3.4 for a detailed description of each step.

## 7.11 Discriminator epoch – Reverberator update procedure

---
**Algorithm 1** Discriminator training epoch on generated data.

---

$R_v^t \leftarrow R_v$         ▷ Initialize target networks with current reverberator network weights.
$R_b^t \leftarrow R_b$
$n \leftarrow$ current epoch #
**for** $\{(G(A_i), V_i)\}$ in $generated\ data\ batches$ **do**
    $\{s_i\} \leftarrow \mathcal{M}(\{(G(A_i), V_i)\})$       ▷ Compute metric scores (using $R_v$ and $R_b$).
    $D \leftarrow \nabla \mathcal{L}_D \Big( D(\{G(A_i)\}), \{s_i\} \Big)$       ▷ Update discriminator.
    $R_v^t \leftarrow \nabla \mathcal{L}_{visual} \Big( R_v^t(\{(G(A_i), V_i)\}), A_i \Big)$   ▷ Update target networks during disc. training.
    $R_b^t \leftarrow \nabla \mathcal{L}_{blind} \Big( R_b^t(\{(G(A_i), V_i)\}), A_i \Big)$
**end for**
**if** $n\%E$ is 0 **then**     ▷ every E epochs, copy target network weights into current reverberators.
    $R_v \leftarrow R_v^t$
    $R_b \leftarrow R_b^t$
**end if**

---

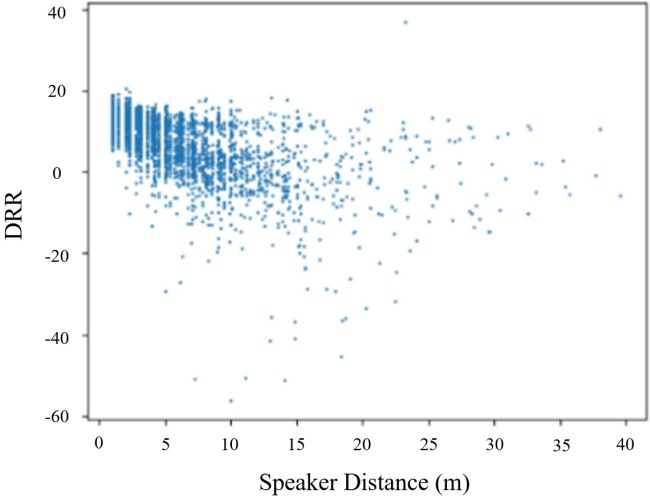

Figure 9: **DRR vs. Speaker distance.** We plot the Direct-to-Reverberant Energy Ratio (DRR) as a function of the distance between the source and receiver for SoundSpaces-Speech audio samples. As the distance between speaker and receiver increases, the listener receives more reverberant sound than direct sound from the speaker, leading to the observed decrease in DRR.

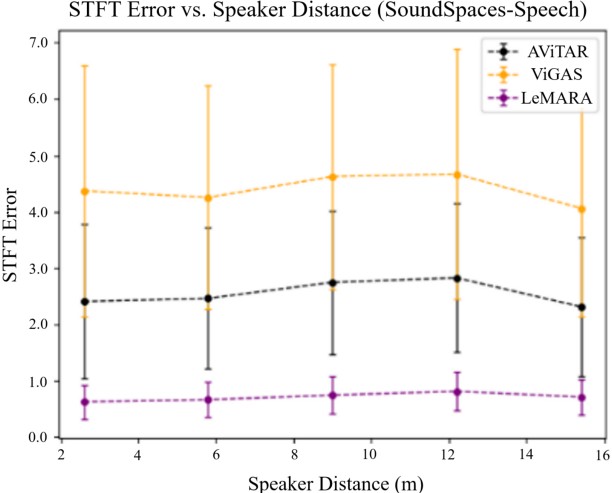

Figure 10: **STFT Error vs. Speaker distance.** We plot STFT Error as a function of the source-receiver distance for SoundSpaces-Speech samples. LeMARA produces the lowest STFT Error at all source-receiver distances.

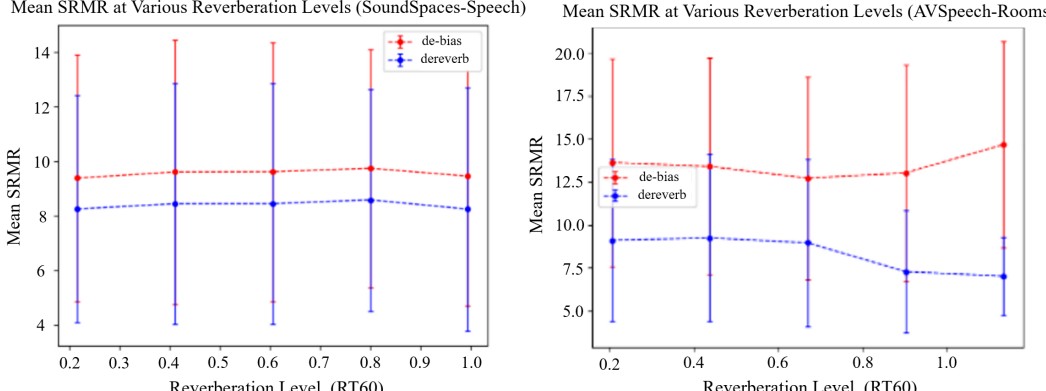

Figure 11: **De-biaser performance at various reverberation levels.** We stratify samples from both datasets by their RT60 and compute the mean SRMR scores of dereverberated and de-biased data within each bin. De-biased audio consistently produces higher SRMR scores than dereverberated audio across all reverberation levels and datasets.

