# OpenReview forum: "Self-Supervised Visual Acoustic Matching"
_NeurIPS.cc/2023/Conference — NeurIPS 2023 poster_

### Official Review · Reviewer_2YZ1 · 2023-07-03

**Soundness:** 3 good
**Presentation:** 2 fair
**Contribution:** 3 good
**Rating:** 5
**Confidence:** 4

**Summary:**

The authors proposed a self-supervised approach to match acoustic conditions via visual information without the need of paired audio-visual data. With this approach, the in-the-wild web data or simulated data can be utilized. They show that this approach can outperforms the state-of-the-art on multiple datasets.

**Strengths:**

- The proposed approach is novel in several aspects, 1. does not require paired audio-visual data, 2. Able to leverage more data from the web or from simulation.
- The authors provide human perception study, which is important for their claims in perception accuracies.
- This technique has potential applications in various audio domains such as domain adaptation, help improving in-the-wild acoustic event classification, etc.

**Weaknesses:**

- The writing of this paper is not easy to follow and missing content, for example in line 200 it refers to Section 4 for details in "RT60 to allow generalization", which does not existing in Section 4, mostly focusing on datasets. In line 226, the off-the-self dereverberator is referred to Section 5, which is difficult to locate for the readers.
- In Section 5, for all the implementation details, suggest the authors to organize and highlight with a table, what data is used to train in which step. Currently it is not easy to follow in the scripts and have clear understanding of the experimental designs.
- The paragraph of line 188 seems to appear in the wrong place, referred Figure 4 is in the results section, while the narrative appears in the approach section.

**Questions:**

- The use of RT60 needs more explanation, is the intuition to model just the acoustic environment and disentangled with the content involved?
- Why is Table 1 on the far right, train with AVSpeech-Rooms and test with LibriSpeech does not include comparisons of STFT?

**Limitations:**

- No potential social or ethical implications.

---

> ### Author Rebuttal · Authors · 2023-08-10
>
> **Thank you for the valuable feedback.**
>
> **Weaknesses:**
>
> **1) The writing of this paper is not easy to follow and missing content, for example in line 200 it refers to Section 4 for details in "RT60 to allow generalization", which does not existing in Section 4, mostly focusing on datasets. In line 226, the off-the-self dereverberator is referred to Section 5, which is difficult to locate for the readers.**
>
> A: Thank you for pointing out the error in line 200; this has been corrected. Line 226 is on page 6, Section 5 (“Experiments”) can be found on the next page. The second paragraph describes the off-the-shelf dereverberator. Clicking on the section reference also directs here.  We respectfully point out that the other 4 reviewers rate our Presentation clarity as Good or Excellent.
>
> **2) It is not easy to follow in the scripts and have clear understanding of the experimental designs. In Section 5, for all the implementation details, suggest the authors to organize and highlight with a table, what data is used to train in which step.**
>
> A: We thank the reviewer for their suggestion. We have added Table 7 in the rebuttal pdf which displays the training steps for each dataset in concise form.
>
> **3) The paragraph of line 188 seems to appear in the wrong place, referred Figure 4 is in the results section, while the narrative appears in the approach section.**
>
> A: We placed this paragraph with references to Figure 4 immediately after introducing the concept of de-biasing in order to help the reader intuitively grasp what the de-biaser is doing to audio visually, before they progress with the rest of the method.
>
> **Questions:**
>
> **1) The use of RT60 needs more explanation, is the intuition to model just the acoustic environment and disentangled with the content involved?**
>
> A: We describe our reasoning for using RT60 in the paragraph at line 194. RT60 is a content-invariant measure of room reverberation, and is a function of room geometry and surface material absorption and reflection properties that characterize a room. These features make RT60 the optimal metric for our task, where we are evaluating whether two waveforms with potentially mismatched content audio sound as if they were recorded in the room.  The metric is used frequently in the literature, and particularly for evaluating acoustic matching quality in prior work [35,23,19].
>
> **2) Why is Table 1 on the far right, train with AVSpeech-Rooms and test with LibriSpeech does not include comparisons of STFT?**
>
> A: STFT is not applicable for that experiment, as explained in Line 291-293.  The LibriSpeech experiment (Table 1, last two columns) performs visual acoustic matching using anechoic source speech samples from the LibriSpeech dataset and real-world target images from AVSpeech-Rooms. Thus the predicted and ground truth reverberant output in this experiment have different speech content, so the STFT metric is not applicable.  Applied here, STFT would capture differences in audio *content* (which by definition must be different here) instead of measuring error in acoustic properties alone. Hence, for this setting we report errors in the reverberant properties of the audio (RT60 error statistics), as well perceptual accuracy via the human user study.

---

> > ### Comment · Reviewer_2YZ1 · 2023-08-18
> >
> > Thank to the authors for your explanations and clarifications. Please include Table 7 from the rebuttal in the final paper if possible. I am going to change the rating.

---

### Official Review · Reviewer_Ky2n · 2023-07-05

**Soundness:** 3 good
**Presentation:** 3 good
**Contribution:** 3 good
**Rating:** 6
**Confidence:** 4

**Summary:**

The paper introduces a self-supervised method for visual acoustic matching (VAM), where the training samples consist of only the target scene image and audio. The paper is well-written and easy to follow. The experimental results have validated the effectiveness of the proposed approach. We have the following comments:
(1) The proposed system is somewhat simple as it incorporates existing methods. Therefore, the scientific depth presented in the paper may not be up to the standards presented at the prestigious conference NeurIPS.
(2) It would be beneficial to include an analysis of the dereverberator. The authors can evaluate the dereverberator's performance using SRMR with different levels of reverberation. It would also be interesting to see the results with and without the De-biaser component based on SRMR.
(3) Please provide results for environments with mild, moderate, and severe reverberant conditions. This will help demonstrate the effectiveness of the LeMARA approach more clearly.
(4) It is crucial to showcase the impact of the De-biaser component. In addition to the spectrograms presented in Figure 4, I recommend that the authors perform additional experiments using supplementary metrics like SRMR or a readily available ASR model. This will provide further evidence and insight into the effectiveness of the De-biaser approach.
(5) It would be informative to report the results of the LeMARA approach separately for seen and unseen images.
(6) Figure 4 shows that LeMARA exhibits greater variation in RT60 compared to AViTAR. Please provide an explanation for this observation.
(7) We suggest preparing an anonymous website where several sets of sound samples of the source, target, AViTAR, and LeMARA can be presented.
(8) Figure 5 indicates that AViTAR outperforms LeMARA in the left-down example. Please provide an explanation for this result.

**Strengths:**

The paper introduces a self-supervised method for visual acoustic matching (VAM), where the training samples consist of only the target scene image and audio. The paper is well-written and easy to follow. The relevance of the research task holds significant value in AR/VR systems and can serve as a crucial element. The experimental results have validated the effectiveness of the proposed approach.

**Weaknesses:**

Some analyses, which can further demonstrate the effectiveness of the proposed approach, are missing, for example:
(1) The authors can evaluate the dereverberator's performance using SRMR with different levels of reverberation. It would also be interesting to see the results with and without the De-biaser component based on SRMR.
(2) Please provide results for environments with mild, moderate, and severe reverberant conditions. This will help demonstrate the effectiveness of the LeMARA approach more clearly.
(3) It is crucial to showcase the impact of the De-biaser component. In addition to the spectrograms presented in Figure 4, I recommend that the authors perform additional experiments using supplementary metrics like SRMR or a readily available ASR model. This will provide further evidence and insight into the effectiveness of the De-biaser approach.
(4) It would be informative to report the results of the LeMARA approach separately for seen and unseen images.
(5) In Line 30, there is one reference missing.
(6) In Fig. 2(b), D(G(A_T)) should be D(G(A_t)).

**Questions:**

(1) A critical point is to showcase the effectiveness of the De-biaser component. Please provide additional experiments to demonstrate its performance.
(2) It s interesting to see the achievable performance of LeMARA approach under mild, moderate, and severe reverberant conditions.
(3) It is crucial to report the results of the LeMARA approach separately for seen and unseen images.

**Limitations:**

(1) The proposed system is somewhat simple as it incorporates existing methods. Therefore, the scientific depth presented in the paper may not be up to the standards presented at the prestigious conference NeurIPS.
(2) Figure 4 shows that LeMARA exhibits greater variation in RT60 compared to AViTAR. Please provide an explanation for this observation.
(3) Figure 5 indicates that AViTAR outperforms LeMARA in the left-down example. Please provide an explanation for this result.

---

> ### Author Rebuttal · Authors · 2023-08-10
>
> **Thank you for the valuable feedback and positive remarks.**
>
> **Weaknesses:**
>
> **1) The authors can evaluate the dereverberator's performance using SRMR with different levels of reverberation. It would also be interesting to see the results with and without the De-biaser component based on SRMR.**
>
> A: Thanks for the suggestion.  This is easy to add to our analysis.  Figure 7 and Table 4 in the rebuttal pdf shows the SRMR score of dereverberated and de-biased data at various levels of reverberation. Overall, the de-biaser produces audio that is significantly cleaner and more anechoic than de-reverberated data.
>
> **2) Please provide results for environments with mild, moderate, and severe reverberant conditions.**
>
> A: Figure 8 in the rebuttal pdf shows Relative RT60 Error across “bins'' of increasing reverberation. Relative RTE generally increases with reverberation across both datasets, likely because highly reverberant audio contains strong residual acoustic clues that may be difficult to completely remove, whereas this is easier in audio that has less reverberation to begin with. On AVSpeech-Rooms and LibriSpeech generalization, LeMARA achieves lower RTE than the baselines even in this high reverberation regime, due to the de-biaser’s ability to strip away even strong residual acoustic clues in audio.
>
> **3) It is crucial to showcase the impact of the De-biaser component.**
>
> A: Our results focus on the correctness of the reverberator outputs (Table 1 in the main paper, Tables 5 and 6 in rebuttal pdf), given that the task is visual acoustic matching, i.e., re-synthesizing reverberation for a new environment.  In particular, rows 3 and 5 in Table 1 of the main paper pinpoint the impact of the de-biaser.  Furthermore, our ablations table in Supp. confirms the effectiveness of our novel de-biaser.  However, we appreciate that the reviewer is interested in understanding how good the “under the hood” de-biaser module is itself.  To that end, Table 4 in the rebuttal pdf highlights the impact of de-biasing on speech quality and reverberation across a variety of metrics, including SRMR. Figure 13 shows the distribution of SRMR scores for reverberant, dereverberated, and de-biased audio. Especially on real-world data (AVSpeech-Rooms), we observe that de-biased data has significantly better quality than dereverberated data.
>
> **4) Informative to report the results of the LeMARA approach separately for seen and unseen images.**
>
> A: Thanks for raising this point. We report results on seen data in Table 6 in the rebuttal pdf. Our LeMARA method outperforms baselines on a variety of RT60 and STFT based metrics across both datasets. Please refer to Table 1 (main paper) and Table 5 (in rebuttal pdf) as well Lines 305-327 for the results and original analysis on unseen data.We reported test-unseen in the main paper as this is the more robust evaluation of performance and due to space constraints.
>
> **5) In Line 30, there is one reference missing.**
> **6)In Fig. 2(b), D(G(A_T)) should be D(G(A_t)).**
>
> Thank you, fixed.
>
> **Questions:**
>
> **1) Critical to showcase the impact of the De-biaser component.**
>
> A: Please see our response for **Weaknesses Question 3**.
>
> **2) Interesting to see the achievable performance of LeMARA approach under mild, moderate, and severe reverberant conditions.**
>
> A: Please see our response for **Weaknesses Question 2**.
>
> **3) Crucial to report the results of the LeMARA approach separately for seen and unseen images.**
>
> A: Please see our response for **Weaknesses Question 4**.
>
> **Limitations:**
>
> **1) Proposed system is somewhat simple as it incorporates existing methods.**
>
> A: Existing approaches for VAM [3] use an off-the-shelf model dereverberator to pre-process speech before acoustic matching. In contrast, we propose a mutual-learning based approach that cyclically optimizes a de-biaser to strip away acoustics and a reverberator to add these acoustics back in. We construct a time-domain GAN model and novel adversarial training objective (Acoustic Residue) which assigns a scalar value to the amount of residual acoustic information in an audio clip. We build a dual-WaveNet model to represent the Acoustic Residue function (see Figure 2c), consisting of a blind WaveNet model and a visual-conditioned WaveNet with RT60 estimation network heads, which are used to compute the Acoustic Residue loss signal. Finally, we also devise a novel training strategy for our GAN (Line 209) that uses the Acoustic Residue signal to not only optimize the generator component as in a traditional GAN, but to also update the Acoustic Residue networks themselves, mitigating the distribution shift that occurs in generated data over the course of GAN training. Please refer to Lines 158, 209, 130, and 120 for details on these novel elements.
>
> **2) Figure 4 shows that LeMARA exhibits greater variation in RT60 compared to AViTAR. Please provide an explanation.**
>
> A: This is an important attribute of our approach.  Both AViTAR [3] and ViGAS [5] perform VAM on dereverberated audio that has residual acoustic reverberation. Thus they learn to add in less reverberation than would be necessary if training on true anechoic data. At test time, however, this leads to under-reverberation, as seen prominently in Figure 11 in the rebuttal pdf.  In contrast, our LeMARA is trained on de-biased audio that has been adversarially optimized to strip away reverberation (Lines 56-59). Given this pseudo-anechoic audio and the natural variation of RT60 values in our training data, the reverberator in LeMARA correctly learns to add a wider variation of reverberation levels into the audio, conditioned on the image.
>
> **3) Figure 5 indicates that AViTAR outperforms LeMARA in the left-down example. Please provide an explanation.**
>
> A: We speculate that the irregular room shape along with camera lens distortion makes the room appear artificially small, leading the model to under-reverberate the audio (Lines 340-342).

---

> > ### Comment · Reviewer_Ky2n · 2023-08-21
> > **Response to rebuttal**
> >
> > The authors have well addressed our concerns. We decided to raise our score.

---

### Official Review · Reviewer_AZuk · 2023-07-06

**Soundness:** 3 good
**Presentation:** 3 good
**Contribution:** 2 fair
**Rating:** 5
**Confidence:** 4

**Summary:**

This paper addresses the task of "visual acoustic matching" (VAM): taking a source audio clip and target visual environment (i.e. an image), and modifying the source audio clip such that it sounds like the clip was recorded in the target environment. The paper proposes a self-supervised approach for training neural networks to solve this task, which can train on examples that only contain the target audio and target visual environment. The idea is to disentangle the room acoustics from the audio content, such that the source room acoustics can be removed, then an audio-visual model can transform this audio to match the target visual environment. The method uses a conditional GAN framework, where an "acoustic residue" metric is defined that measures the discrepancy between audio-only and audio-visual reverberation. The acoustic residue metric is used to train a discriminator that can be used to train a "de-biaser" generator model that strips additional reverberation from an initially dereverberated signal. The output of the de-biaser can then be fed to the audio-visual reverberator.

Although the method can theoretically be applied to other types of signals, the paper focuses on clean speech as the audio signals. Datasets used are SoundSpaces-Speech (reverb impulse responses (RIRs) simulated from Matterport3D scans of homes with 3D-rendered human at source location and audio from LibriSpeech) and AVSpeech-Rooms (YouTube videos that mostly feature a single speaker with little background noise). Objective metrics are used to measure performance: MSE between magnitude spectrograms of predicted and ground-truth speech, and MSE between RT60 estimates of predicted and ground-truth speech. The proposed method achieves better objective scores compared to comparable methods. A perceptual study is also done, which shows users could identify the room from the proposed method 46.1% of the time, versus 34.7% for a baseline method.

**Strengths:**

**S1)** The method provides a means of training self-supervised models for VAM.

**S2)** Evaluation includes a human perceptual study.

**S3)** Demo video is clear and helps with understanding the method. Also, thanks for providing audio demos, they are very useful for evaluating the method and how it compares to the AViTAR baseline.

**Weaknesses:**

**W1)** The evaluation of the method is weak. Watching the demo video, it seems like this is certainly a difficult task to evaluate, since the effect can be rather subtle. Nevertheless, I think the evaluation metrics could be improved.

First, there are better options than just measuring MSE on magnitude spectrograms. Human hearing is logarithmic, so MSE on linear magnitude is a poor match to human perception. An easy alternative is to measure MSE between log spectrograms, although this encounters an issue with where to set the floor on the log to avoid -infinity. A solution to this is to use magnitude raised to a power (e.g. 0.3), which approximates the log, but also goes to 0 where the magnitude is 0. This may help this this issue: "Models that use this dereverberator without further de-biasing will display artificially low STFT error when evaluated in-dataset"

Second, there are other important properties of reverb impulse responses besides RT60, such as direct-to-reverberant ratio (DRR). DRR measures the ratio of energy of the direct path to the energy of the reverberant part, and can be an important cue for distance of the source. Also, distance of the source is not really discussed, and DRR is a crucial property to help measure this (see W3).

Third, the comparison of the RT60 distributions in Figure 4 are not very convincing, because they are just box plots, and it's not clear how the proposed approach "more closely matches the ground truth target distribution.": the median is closer to ground-truth than the baseline, but it seems like a different type of plot would allow more detailed comparison of the distributions, like histograms or violin plots.

**W2)** The paper restricts its focus to a clean single speaker, which suggests the method may be limited on real-world data, which can contain a great variety of non-speech sounds, and also multiple speakers. I don't think the proposed method could be applied directly to these scenarios, because all sounds would need to first be separated, then each processed by the pipeline (i.e. each sound dereverberated and re-reverberated)

**W3)** The paper does not discuss the effect of distance of an object, nor the effect of distance on the reverberation. In particular, the DRR is an important property of reverberation that is totally ignored in this paper. At the very least, assumptions about how the method handles distance and location of sources needs to be clearly described and discussed, and also I think evaluation metrics should take distance into account (see W1).

**Minor comments and typos**

a) "data, we" -> "data, and we"

b) "target space [?,..": update missing ref

c) "In-the-wild Web" -> "In-the-wild web"

d) "We focus on human speech in indoor settings": clearly specify that it's clean speech, without background noise

e) "this leaves signals of the target environment": I think "residuals" would be better than "signals"

f) "Unlike SRMR, DNSMOS [31], or any existing off-the-shelf metric that quantifies dereverberation,": I don't think DNS-MOS is trained to quantify reverberation specifically, it's overall quality, which can be affected by other properties such as background noise and/or artifacts. May be good to adjust this description

**Questions:**

Q1) Looking at the spectrograms of de-biased audio in Figure 4 and listening in the demo video, de-biased audio seems like it is applying fairly aggressive suppression, perhaps removing some energy of the anechoic audio. This seems like a weakness of the method. Are there mechanisms to prevent over-suppression of the anechoic audio? Does this suggest a trade-off for the de-biaser between dereverberating and suppressing signal? Some discussion of this would be helpful.

Q2) In Figure 2, where does "A" come from (lower right panel)? I guess this is the de-biased audio? Would be good to make this more clear

**Limitations:**

Limitations described adequately.

---

> ### Author Rebuttal · Authors · 2023-08-10
>
> **Thank you for the valuable feedback and questions.  We hope that our clarifications about the dataset contents and incorporation of the additional suggested metrics help reconsider our contribution.**
>
> **Weaknesses:**
>
> **W1)**
>
> **I) There are better options than just measuring MSE on magnitude spectrograms. An easy alternative is to measure MSE between log spectrograms**
>
> A: Thank you for this suggestion.  We have added in log magnitude STFT loss as an additional metric in Tables 5, and 6 in the rebuttal pdf. We again outperform the current SOTA approach (AViTAR [3]) on this metric across both datasets.
>
> **II) There are other important properties of reverb impulse responses besides RT60, such as direct-to-reverberant ratio (DRR). …does not discuss the effect of distance…**
>
> We discuss the speaker-position dependent nature of RIRs and how this is addressed in SoundSpaces-Speech data creation at Lines 247-255.
>
> DRR estimation from reverberant audio is challenging. Spectrograms mix both direct and reverberant sound components across frequency and time, which makes it difficult to estimate DRR even if pre-training the model on synthetic audio. Spectrogram-based RT60 estimation is more reliable, as it can be accurately estimated from energy reduction across temporal bins on the spectrogram.  If we have misunderstood the suggestion, we’d welcome a pointer on how to allow DRR as a reliable metric when one lacks ground truth RIRs.
>
> To illustrate the effect of speaker distance on performance, we provide an STFT Error vs speaker distance plot, Figure 16 in the rebuttal pdf (for SoundSpaces, where we have GT RIRs). Overall, we observe minimal change in error as a function of speaker distance. To provide an analysis of DRR on our data, Figure 15 displays the relationship between speaker-distance and DRR on samples from SoundSpaces-Speech. We observe that DRR generally decreases as a function of speaker distance.
>
> **III) RT60 distributions in Figure 4 are not very convincing, because they are just box plots…a different type of plot would allow more detailed comparison of the distributions, like histograms or violin plots.**
>
> A: Thank you for the valuable suggestion for how to best present the output distributions.  We now provide histogram and violin plots of the source, predicted, and target RT60 distributions (Figures 9-12 in rebuttal pdf).  We believe this is indeed a more direct way to see that the proposed approach more closely matches the target distribution on all three setups, compared to our original boxplots.
>
> **W2)**
>
>  **The paper restricts focus to a clean single speaker, which suggests the method may be limited on real-world data, which can contain a great variety of non-speech sounds, and also multiple speakers.**
>
>  A: Many real-world videos consist of a single speaker (instructional videos, presentations, vlogs).The AVSpeech dataset from which our training set is sampled consists of over 290k such videos from YouTube. Although these videos are single speaker, we disagree with the characterization as little background noise.  The clips contain a variety of non-speech background sounds (e.g. white noise from air conditioning, clicking/tapping noises from object interactions, background music).  Our model performs well in this real-world setting, as shown by the results on AVSpeech-Rooms in Tables 1 (main paper), 4,5, and 6 (rebuttal pdf) in which we consistently outperform the SOTA.
>
> Conceptually our approach does not make any assumptions about the type of audio—whether speech or single speaker (see Line 67, footnote 2). Given a dereverberator and RT60 estimator trained on the relevant types of sounds (mixtures of speakers and/or non-speech sounds), our approach can be used to train a VAM model directly on these mixtures. In principle, audio source separation for sources close by in the same environment would not be necessary; they would all be influenced by the same room acoustics.  We leave such explorations for future work.
>
> **W3)**
>
> **The paper does not discuss the effect of distance of an object, nor the effect of distance on the reverberation...(see W1)**
>
> A: Please refer to our earlier response in **W1 part II**.
>
> **Minor comments and typos**
>
> A: Thank you, fixed.
>
>
> **Questions:**
>
> **Q1) Are there mechanisms to prevent over-suppression of the anechoic audio? Does this suggest a trade-off for the de-biaser between dereverberating and suppressing signal? Some discussion would be helpful.**
>
> A: Thank you for the insightful questions. The de-biaser may remove some energy from audio, but the spectrogram-based loss provides a strong signal for the reverberator to add this energy back during fine-tuning. This can be seen in the predicted reverberant audio spectrograms in the demo video. There does exist a trade off between dereverberation and suppression: if the de-biaser's aggressive dereverberation also removes energy from audio, then the reverberator fine-tuned on this data may learn to add more energy back than is necessary, which at test time may hurt STFT error. Optimizing de-biased speech for SRMR helps mitigate this, preserving low frequency modulation energy - speech content - over high frequency modulation energy (reverberant content). Pre-training with SRMR provides the generator with a strong prior for preserving energy associated with speech content in de-biased audio, guarding against indiscriminate energy suppression during Acoustic Residue fine-tuning. Also, energy removal may contribute to larger STFT based error, but it does not affect reverberant properties, which our human user study shows may be more important in evaluating perceptual accuracy (Line 343).
>
> **Q2) In Figure 2, where does "A" come from (lower right panel)? I guess this is the de-biased audio? Would be good to make this more clear.**
>
> A: “A” is the input to the metric, which can be either the de-biased audio or the reverberant audio. We have updated the figure to more clearly show this.

---

> > ### Comment · Reviewer_AZuk · 2023-08-19
> > **Response to rebuttal**
> >
> > Thanks to the authors for the detailed responses and rebuttal PDF.
> >
> > > If we have misunderstood the suggestion, we’d welcome a pointer on how to allow DRR as a reliable metric when one lacks ground truth RIRs.
> >
> > I guess I was thinking of blind DRR estimation, which is certainly not perfect, but could provide some insight. A number of blind DRR methods have been proposed that could be used, even from a single microphone. There is this classic Matlab implementation:
> >
> > https://www.mathworks.com/matlabcentral/fileexchange/32752-blind-direct-to-reverberant-energy-ratio-drr-estimation
> >
> > And here is a more recent paper on blind DRR estimation:
> >
> > https://www.isca-speech.org/archive_v0/Interspeech_2020/pdfs/2171.pdf
> >
> > But in any case, I appreciate the additional results on DRR versus distance.
> >
> > > We now provide histogram and violin plots of the source, predicted, and target RT60 distributions (Figures 9-12 in rebuttal pdf)
> >
> > Thanks, that's definitely an improvement.
> >
> > > We have added in log magnitude STFT loss as an additional metric in Tables 5, and 6 in the rebuttal pdf.
> >
> > I would still encourage the authors to consider MSE on magnitude spectrograms raised to the 0.3 power, i.e. $|X|^{0.3}$.
> >
> > The authors have addressed most of my concerns, so I am willing to raise my score.

---

### Official Review · Reviewer_bz7J · 2023-07-06

**Soundness:** 3 good
**Presentation:** 4 excellent
**Contribution:** 3 good
**Rating:** 6
**Confidence:** 4

**Summary:**

This work proposes a method for visual acoustic matching, the task of processing an audio signal so the room acoustics are perceived as originating from within a certain room based on an image. While paired data is generally required for this task, the proposed approach is self-supervised and trained using “in-the-wild” videos of speakers in various rooms. To achieve this, a dereverberation system along with a de-biaser, a visually-conditioned reverberator, and a blind reverberator are used. The de-biaser is trained by optimizing a novel acoustic residue metric that measures the relative difference between the amount of acoustic information in two recordings. While similar to dereverberation, this de-biasing process can be seen as post-processing step that aims to further reduce information in the dereverberated output so the visually conditioned reverberator model must use cues from the supplied image to match the target room acoustics. To use this acoustic residue metric as an objective function, the de-biaser is trained in a MetricGAN fashion where the discriminator is trained to regress the true metric value. Two evaluation datasets are used and results indicate that the proposed method outperforms existing baseline systems. In addition, a human perception study also validates these results.


**Strengths:**


1. The proposed method is original and focuses on combining multiple existing approaches along with a novel acoustic residue metric in order to achieve a system that can be trained without paired data. These approaches include the use of various pretrained submodules, including a dereverberation model and a RT60 estimation model, as well as the MetricGAN approach that enables the use of the acoustic residue metric during optimization.

2. As outlined by the authors, existing approaches are often greatly limited by the availability of paired data. Not only is the proposed approach capable of being trained without paired data, the evaluation indicates superior performance to existing methods.


**Weaknesses:**

1. Some relevant references for work in audio-only acoustic matching are omitted. Works such as [1] are similar in nature but rely on conditioning from an audio signal instead of an image. Major components of the reverberator architecture are similar as well, such as the use of a WaveNet architecture for audio processing. This would be highly relevant in the context of the blind reverberator in this work. Also, other audio only approaches such as [2] would be relevant to include in the context of related work. It would also be relevant to address other GAN-based reverberation models such as [3, 4].

2. While overall the manuscript is well organized and clear, further refinement with regards to the explanation of the process for training the complete system may help reduce the burden for readers. For example, perhaps an algorithm listing each step of the process and referencing the variables used in the text would be helpful.

3. While the evaluation considers two datasets and appears relatively sound, it is still somewhat difficult to understand the relative improvement afforded by the proposed method in comparison to the baselines.

4. While the authors are commended for including a human perception study, the design of the study may be somewhat flawed, making it difficult to make any conclusion from the results.


[1]	Su, Jiaqi, Zeyu Jin, and Adam Finkelstein. "Acoustic matching by embedding impulse responses." ICASSP 2020-2020 IEEE International Conference on Acoustics, Speech and Signal Processing (ICASSP). IEEE, 2020.

[2]	Steinmetz, Christian J., Vamsi Krishna Ithapu, and Paul Calamia. "Filtered noise shaping for time domain room impulse response estimation from reverberant speech." 2021 IEEE Workshop on Applications of Signal Processing to Audio and Acoustics (WASPAA). IEEE, 2021.

[3]	Ratnarajah, Anton, Zhenyu Tang, and Dinesh Manocha. "IR-GAN: room impulse response generator for speech augmentation." INTERSPEECH (2021).

[4]	Ratnarajah, Anton, Zhenyu Tang, and Dinesh Manocha. "Ts-rir: Translated synthetic room impulse responses for speech augmentation." 2021 IEEE Automatic Speech Recognition and Understanding Workshop (ASRU). IEEE, 2021.


**Questions:**

Is it fair to call the proposed approach “self-supervised”? While the approach does not require paired data directly, it appears that paired data is indirectly required since pretrained models, such as the RT60 estimator and dereverberation models. In other words, it appears that it would not be possible to train the proposed system in a fully self-supervised manner from scratch, without these pretrained modules, which in fact use labeled/paired data. Instead, it may be more accurate to label the proposed method as one capable of using an “unpaired” dataset instead of self-supervised. Perhaps terms such as “semi-supervised” or “weakly-supervised” are more appropriate?

What is the motivation for using the MetricGAN approach to construct a differentiable proxy of the acoustic residue metric $\mathcal{M}$? Based on the text, it appears that the RT60 estimation model $\mathcal{RT}$ is implemented with a neural network and hence differentiable. While (3) involves non-differentiable functions such as the absolute value and $\max$, these are both approximately differentiably, indicating that it may be possible to directly backprop through the metric $\mathcal{M}$. Is this something the authors considered? If so, was the failure of this approach the motivation to use the MetricGAN approach? Some discussion on the motivation for the MetricGAN approach would be beneficial.


Line 30: A reference has not been rendered properly “?”


**Limitations:**

It could be beneficial to include not only the mean RT60 error (RTE (s) as reported in Table 1, but also the bias and correlation coefficient, which is common practice in RT60 estimation evaluation. This can help to establish how the model is failing especially if the test dataset has a non uniform distribution of RT60 targets.

One limitation of the use of the RT60 estimation in the proposed acoustic residue metric is that this measurement appears to consider only the broadband RT60. It is generally accepted that in most rooms the RT60 can vary quite significantly across frequency, for example, with the higher frequencies decaying at a significantly faster rate than the low frequencies (generally due to absorption from the material properties). As a result, this variation of RT60 across frequency often provides significant insight into the character of the room, which is often detectable by a human listener. Therefore, using only the broadband RT60 in the acoustic residue metric may lead to some cues about the bandwise RT60 remaining in the de-biased recording as long as the broadband RT60 is perturbed. Future work could consider a band-wise RT60 estimation.

---

> ### Author Rebuttal · Authors · 2023-08-10
>
> **Thank you for the valuable feedback, insights, and positive remarks.**
>
> **Weaknesses:**
>
> 1) **Some relevant references for work in audio-only acoustic matching are omitted.**
>
> A: Thank you for bringing these audio-only works to our attention.  We are happy to cite them. Our discussion of audio-only acoustic matching is in Lines 30-33, and on Line 79 including NAF ([21]). Our discussion on GAN-based enhancement models is in Lines 94-102 including Sergan ([1]) and MetricGAN models ([11,12,13]).
>
> (Su et al., ICASSP 2020) is relevant audio-only acoustic matching work, which shares our general conditional WaveNet architecture as does [38,32] and the Acoustic Synthesis module of [5] (i.e., it is not unique to (Su et al., ICASSP 2020)). (Su et al., ICASSP 2020) uses training data in which the same utterances are recorded in different environments, motivating an approach that is quite different from our setting, where every (utterance, environment) pair is unique. This key difference motivates the rest of our architectural innovation (adversarial de-biaser, Acoustic Residue metric, dual-WaveNet reverberator, and mutual learning framework).
>
> (Steinmetz et al., IEEE 2021)  addresses the related task of RIR generation along the lines of [35,23]. (Ratnarajah, INTERSPEECH 2021) and (Ratnarajah, IEEE 2021) develop approaches for this same task using conventional image-based GANs. [35,23,11,12,13] collectively address RIR generation and audio-based GAN methods which these two works can be grouped with.
>
> 2) **While overall the manuscript is well organized and clear, provide refinement ... perhaps an algorithm listing each step.**
>
> A: Thank you for this suggestion.  We have added an algorithm box in the rebuttal pdf, and a tabular description of our three stage training process. (Figure 14, Table 7).
>
> 3) **While the evaluation considers two datasets and appears relatively sound, it is still somewhat difficult to understand the relative improvement afforded by the proposed method in comparison to the baselines.**
>
> A: Table 1 highlights the quantitative improvement in our method over baselines in two datasets using standard metrics (and see Tables 5 and 6 in the rebuttal pdf for additional metrics). We also provided a human user study showing our method outperforms the SOTA (line 343).  Additionally, our demo video (Supp.) contains examples comparing audio generated by our model and the current SOTA (AViTAR [3]) for reviewers to better understand the perceptual improvement.  We believe that this mix of both hard metrics on the one hand and perceptual studies and qualitative examples on the other hand facilitate understanding the relative improvement.  If there is a more specific question from the reviewer, we’d be happy to address it.
>
> 4) **While the authors are commended for including a human perception study, the design of the study may be somewhat flawed.**
>
> A: The setup details start at line 343. We believe this is a solid perception study following good practices. However, if we have missed anything, could the reviewer please indicate exactly what might be somewhat flawed?
>
> **Questions:**
>
> 1) **Is it fair to call the proposed approach “self-supervised”? Perhaps terms such as “semi-supervised” or “weakly-supervised” are more appropriate?**
>
> A: We appreciate this suggestion and we agree there’s room for different interpretations of the terms.  To explain our thinking: While the RT60 estimator and off-the-shelf dereverberator require supervision for their training, they are treated as frozen modules in our method. As such, our model can be extended to a new dataset with no additional supervision necessary, as we demonstrated with the application to AV-SpeechRooms. Terms such as “semi-supervised” and “weakly-supervised” imply that the amount of paired data required scales with the amount of full training data, whereas our method requires only the finite amount of training data on which the RT60 estimator and dereverberator modules have been trained. However we appreciate the reviewer’s point, and “unpaired” may be a more appropriate term than “self-supervised” given this ambiguity.
>
> 2) **Q: What is the motivation for using the MetricGAN approach to construct a differentiable proxy of the acoustic residue metric? Some discussion would be beneficial.**
>
> A: Thanks for the insightful question.  While the Acoustic Residue metric is approximately differentiable, our metric optimizes for both Acoustic Residue as well as SRMR (for both reverberation and speech quality), and the implementation of SRMR we used is not differentiable, motivating the use of the MetricGAN. This balance is important as we found that incorporating SRMR into the metric improves performance beyond a pure Acoustic Residue metric (See ablations table in Supp.), and provides stability during reverberator fine-tuning (lines 232-237). Our approach also allows for extensibility to other non-differentiable speech quality scores such as PESQ.
>
> 3) **Q: Line 30: A reference has not been rendered properly “?”**
>
>  A: Thank you.
>
> **Limitations:**
>
> 1) **It could be beneficial to include not only the mean RT60 error (RTE (s) as reported in Table 1, but also the bias and correlation coefficient.**
>
> A: Thank you for the suggestion. In short, we’ve added the requested error metrics and they reinforce our original claims. See Tables 4, 5 and 6 in the rebuttal pdf.
>
> 2) **One limitation of the use of the RT60 estimation in the proposed acoustic residue metric is that this measurement appears to consider only the broadband RT60. Future work could consider a band-wise RT60 estimation.**
>
> A: We thank the reviewer for their insight.  While our results already show consistent gains using the broadband RT60 in the residue metric, we agree it will be interesting future work to explore a band-wise variation.

---

> > ### Comment · Reviewer_bz7J · 2023-08-18
> >
> > Thank you for the clarification and additional details provided in the rebuttal document. These results reinforcement the claims made in the paper surrounding the superiority of the proposed approach.
> >
> > As previously argued, the term "self-supervised" is probably not the most accurate due to the required pre-training of the supporting models, which use supervised data. However, this point should not limit the acceptance of this work. I will retain my original score and advocate for acceptance.

---

### Official Review · Reviewer_7iLe · 2023-07-10

**Soundness:** 3 good
**Presentation:** 3 good
**Contribution:** 3 good
**Rating:** 6
**Confidence:** 4

**Summary:**

This work aims to perform acoustic matching with unpaired training data, i.e. observing the audio only in the target environment without samples of the same audio in the source environment. The basic process involves the common process of dereverberator and reverberator. But one key point in this work is to introduce an acoustic residue metric to measure the residual information in a waveform to ensure the quality of dereverberated audio. Finally, experiments on SoundSpaces-Speech and AV Speech show its effectiveness.

**Strengths:**

The presentation is very clear. The motivation to tackle the task is also intuitive and meaningful. The method is reasonable. I like this smart and simple manner proposed in this paper to tackle the key limitation in visual acoustic matching.

**Weaknesses:**


(1) Is it limited to evaluate with only two metrics, STFT and RTE? (2) Although the number of methods using unpaired data may be few, the number of methods using paired data should be large. There could be some comparison with them to show the performance level of the proposed method. For example, comparing on other paired datasets with existing methods which use paired data could show the difference of the methods use and not use paired data.

**Questions:**

(1) The error of STFT changes obviously in AVSpeech-Rooms compared with other settings, i.e. about 6~7 vs 1~2, what’s the possible reason? I t would be nice to give the clarification. (2) Two questions about the evaluation metrics and the comparison are as listed in weakness.

**Limitations:**

yes.

---

> ### Author Rebuttal · Authors · 2023-08-10
>
> **Thank you for the valuable feedback and positive remarks.**
>
> 1) **Is it limited to evaluate with only two metrics, STFT and RTE?**
>
> A: We focus on RT60 and Spectrogram loss metrics as these capture well the reverberant acoustic properties of audio, and are used in prior works for acoustic matching [3, 23, 35]. Furthermore, we also provide quantitative results of a human user study for evaluation of perceptual quality in the paragraph starting at line 343.  Based on the reviewers’ helpful suggestions of additional metrics, we now also provide log magnitude STFT error, RT60 bias/correlation coefficient, and relative RT60 error (Tables 5 and 6 in the rebuttal pdf). We also evaluate the de-biaser and dereverberator using a speech quality metric (SRMR) (Table 4 in the rebuttal pdf). These new metrics continue to support our claims, especially on real-world data.
>
> 2) **Although the number of methods using unpaired data may be few, the number of methods using paired data should be large. There could be some comparison with them to show the performance level of the proposed method. For example, comparing on other paired datasets with existing methods which use paired data could show the difference of the methods use and not use paired data.**
>
> A: The task of Visual Acoustic Matching (VAM), first introduced in [3], is relatively new, and as such there are very few existing methods, whether using either paired or unpaired data. To the best of our knowledge, we have compared our method against all available relevant works: the original and only Visual Acoustic Matching paper [3], as well as ViGAS [5], a model designed for the related task of novel-view acoustic synthesis that we adapt to for VAM (paragraph at line 294). We also evaluate audio-only variants of these models for comparison against our audio-only model. In addition to these, we evaluate a naive approach in which the dereverberated input is copied to the output. Table 1 displays these baselines in full. If the reviewer has a specific baseline in mind that we have not covered, we would welcome the suggestion and are happy to add it.
>
> 3) **The error of STFT changes obviously in AVSpeech-Rooms compared with other settings, i.e. about 67 vs 12, what’s the possible reason? It would be nice to give the clarification.**
>
> A: AVSpeech-Rooms contains a variety of non-speech sounds (e.g. white noise from air conditioning, clicking/tapping noises from object interactions, and even background music) which make proper reverberation even more challenging. These signals may be perceptually weak, but they will show up on a spectrogram and contribute to the larger STFT error which we observe on AVSpeech-Rooms for all methods (Table 1). In contrast, data in SoundSpaces-Speech (derived from a state-of-the-art acoustics simulator) convolves anechoic audio with a Room Impulse Response based on the room geometry (lines 245-252) to produce audio that contains no artifacts or background noises to contribute to spectrogram errors.

---

> > ### Comment · Reviewer_7iLe · 2023-08-18
> >
> >  Thank you for the detailed explanation provided by the authors. I suggest the authors include the comparative results under other metrics mentioned in Q1 in the appendix, to further enhance the persuasiveness of the results. I don't have any other questions, and I still think this is a good piece of work, so I will maintain my score.

---

### Author Rebuttal · Authors · 2023-08-10

Thanks to all reviewers for their time and valuable feedback.

Three reviewers recommend accepting.  The other two suggest additional error metrics of interest and ideas for how we plot the results (Reviewer AZuk) and easy-to-address clarifications and items addressed already in the text (Reviewer 2YZ1).  We address all items below and show the requested metrics in the rebuttal pdf.

---

### Decision · Program_Chairs · 2023-09-21

**Decision:**

Accept (poster)

**Comment:**

All reviewers recommend acceptance. The AC sees no basis to overturn the reviews, and thus recommends acceptance. Authors should attend to main points in the reviews, such as discussing suppression (Reviewer AZuk) and discussion of the evaluation metrics (Reviewer 2YZ1), when preparing a final version.